# The Current and Potential Distribution of Parthenium Weed and Its Biological Control Agent in Pakistan

**DOI:** 10.3390/plants12061381

**Published:** 2023-03-20

**Authors:** Asad Shabbir, Myron P. Zalucki, Kunjithapatham Dhileepan, Naeem Khan, Steve W. Adkins

**Affiliations:** 1Weeds Research Unit, Invasive Species Biosecurity, New South Wales Department of Primary Industries, Orange, NSW 2800, Australia; 2School of Agriculture & Food Sciences, The University of Queensland, Gatton, QLD 4343, Australia; 3School of Biological Sciences, The University of Queensland, St Lucia, Brisbane, QLD 4072, Australia; 4Biosecurity Queensland, Department of Agriculture, Fisheries and Forestry, Ecosciences Precinct, Dutton Park, Brisbane, QLD 4102, Australia; 5Department of Weed Science and Botany, The University of Agriculture, Peshawar 25000, Pakistan

**Keywords:** climatic suitability, CLIMEX, *Parthenium hysterophorus*, leaf-feeding beetle, biological control

## Abstract

*Parthenium hysterophorus* L. (Asteraceae), commonly known as parthenium weed, is a highly invasive weed spreading rapidly from northern to southern parts of Pakistan. The persistence of parthenium weed in the hot and dry southern districts suggests that the weed can survive under more extreme conditions than previously thought. The development of a CLIMEX distribution model, which considered this increased tolerance to drier and warmer conditions, predicted that the weed could still spread to many other parts of Pakistan as well as to other regions of south Asia. This CLIMEX model satisfied the present distribution of parthenium weed within Pakistan. When an irrigation scenario was added to the CLIMEX program, more parts of the southern districts of Pakistan (Indus River basin) became suitable for parthenium weed growth, as well as the growth of its biological control agent, *Zygogramma bicolorata* Pallister. This expansion from the initially predicted range was due to irrigation producing extra moisture to support its establishment. In addition to the weed moving south in Pakistan due to irrigation, it will also move north due to temperature increases. The CLIMEX model indicated that there are many more areas within South Asia that are suitable for parthenium weed growth, both under the present and a future climate scenario. Most of the south-western and north-eastern parts of Afghanistan are suitable under the current climate, but more areas are likely to become suitable under climate change scenarios. Under climate change, the suitability of southern parts of Pakistan is likely to decrease.

## 1. Introduction

*Parthenium hysterophorus* L., commonly known as parthenium weed, is an invasive species of global significance and is now found in more than 50 countries worldwide [1,2]. Parthenium weed is native to Mexico, Central America, and parts of South America. Within the last few decades, this weed has expanded its geographic range and is threatening to invade much larger areas of the world in the future [3,4]. Parthenium weed is known to reduce natural biodiversity and agricultural productivity and has negative effects on human and animal health [5]. In South Asia, parthenium weed has invaded all countries and is spreading fast within the region [2,6,7]. In Pakistan, the first record of parthenium weed was reported in the Gujrat district of the Punjab province in 1980 [8]. It is commonly believed that the weed had moved from India to Pakistan through trade and transport between both countries via the Wagah border. The weed has now rapidly spread throughout the Punjab province (Figure 1) as well as to Khyber Pakhtunkhwa (KP), the Islamabad Capital Territory (ICT), and Azad Jammu and Kashmir [9,10]. The core infestations of parthenium weed are in the central and northern districts of Punjab, the ICT, and KP, and it is moving to the southern parts of Punjab and north-eastern KP [6]. However, given parthenium weed’s highly invasive nature, and the existence of extensive irrigated farming in the southern parts of Punjab, it is hypothesized that the weed could and has probably already spread into those southern districts of this province, and possibly to the Sind and Baluchistan provinces in the south and south-west regions of Pakistan [6,9].

The local adaptability of invasive species like parthenium weed to a wide range of environmental conditions, and thus to more diverse habitats, is an important factor in their successful invasion [5,11,12]. One approach to the prevention of invasion by invasive species is to use information on their ecological requirements, and other factors that influence their spread and establishment in new habitats. For instance, climatic similarity between native and introduced ranges of an invasive species has been successfully used to predict the potential distribution of a newly invading plant species. The CLIMEX software package [13] is an important tool that has been used for exploring the effect of the climate on the distribution of arthropod pests, diseases, and weeds [3,14]. This package has successfully been used to predict the potential future distribution of invasive plants under both current and future climate change scenarios [15,16,17], as well as for biological control agents [18,19]. McConnachie et al. [3], using the CLIMEX modeling tool, predicted that parthenium weed could expand its current distribution range in Africa and potentially invade much larger areas of the world. However, the model was somewhat conservative in predicting the suitability of southern Pakistan for parthenium weed. One possible reason for this could have been that the weed is more tolerant to higher temperatures and drier conditions than was entered into the model. In addition, extra moisture could be available for growth that is not provided by the rainfall feature in the CLIMEX model alone. This could be the case in Pakistan as the Indus River basin is irrigated each year, effectively applying more moisture to this region than would have been predicted from rainfall alone. 

Biological control agents for weeds with wide climatic tolerances need to be chosen carefully because specialist herbivores and plant pathogens may have narrower climatic requirements than their hosts. Climatic matching, based on modeling, has been used to predict the establishment and distribution of invasive plants in new areas and then to match the subsequent range of their biological control agents [20]. A leaf-feeding beetle, *Zygogramma bicolorata* Pallister (Coleoptera: Chrysomelidae), from Mexico was tested and released into India in 1984 as a biological control agent against parthenium weed [21]. Since then, this agent has spread to several states in India [7] and into the neighboring countries of Nepal, Bhutan, and Pakistan [22,23]. In Pakistan, *Z. bicolorata* was first found in a forest reserve near Lahore, Punjab province, in 2003 [23]. Presumably, the agent arrived in Pakistan following its release in India and its subsequent dispersal. *Zygogramma bicolorata* is well-established in the northern parts of the Punjab province, ICT, and parts of KP [9,24] To date, there is very little data on the potential distribution of parthenium weed and its biological control agent, *Z. bicolorata*, in Pakistan or on how the future distributions may be affected by climate change.

The aims of the current study were firstly, to develop a predictive distribution of parthenium weed and *Z. bicolorata* in Pakistan for the current and future climate (+3 °C), using a CLIMEX modeling approach. Secondly, we predicted how irrigation might also affect the current and potential future distribution of weed and its biological control agent in Pakistan.

## 2. Results

### 2.1. Parthenium hysterophorus

#### 2.1.1. Potential Range under the Current and Future Climate

The CLIMEX model predicted that most regions of South Asia are climatically suitable for the growth of parthenium weed. This model demonstrates that the whole of Bangladesh, Sri Lanka, India, Nepal, and Bhutan, and the northern regions of Pakistan are all highly favorable for the growth of the weed (Figure 2A). The model also predicts that the south-western part of Afghanistan is moderately suitable for the growth of parthenium weed. The predictive model run to simulate climate change (+3 °C) indicates that the weed could expand further into the northern regions of India, Pakistan, Nepal, and Bhutan, while the southern regions of these countries would become less suitable (Figure 2B,C). More areas within the south and northwest of Afghanistan would become climatically suitable for parthenium weed growth under a changing climate (Figure 2C).

The predictive distribution model operating under the current climate (Figure 3) suggests that the northern districts of the Punjab province (including Narowal, Sialkot, Gujarat, Jhelum, Rawalpindi, and Attock), most of ICT and parts of the Azad Jammu and Kashmir province (AJK), the northwestern districts of the KP (including the Swat, Abbottabad, Mansehra, Haripur, Shangla, Dir, Charsadda, Mardan, Malakand, and Bunair districts), and the Mohmand Agency of the former Federally Administered Tribal Areas have a climate suitable for the growth of parthenium weed, with an average ecoclimatic index (EI) of >30 in these areas (Figure 3A). The most northern regions of Pakistan, such as Gilgit Baltistan and the north-eastern parts of KP and AJK seem to be too cold to sustain parthenium weed growth now. In general, the EI values for parthenium weed decrease from the northern to the southern regions of the Punjab province, as well as in the Baluchistan and Sind provinces (Figure 3B). In a predictive model run to simulate climate change (+3 °C), the projection suggests that parthenium weed may be able to expand its invasion beyond limits further north, while contracting its distribution in the southern extremities of the Punjab and Baluchistan provinces (Figure 3B,C). The AJK and KP will become more suitable for the weed as suggested by an increase in the EI value in these provinces (Figure 3C).

#### 2.1.2. Prediction under Current Climate and under Climate Change with Irrigation

In the Punjab province, there is an extensive canal network carrying water from the Indus River system into the cropping regions where it is used to irrigate the land (Figure 4). This extensive network of canals adds an extra ca. 250 million m^3^ of water annually and this is on top of the regional rainfall. The Indus River system comprises five major rivers (viz. the Jhelum, Chinab, Ravi, Sutluj, and the Indus). When the conditions 0.5 mm day^−1^ (winter) and 1.0 mm day^−1^ (summer) of irrigation water were added to the basic model, the EI values for those regions greatly increased, and the entire basin became favorable for parthenium weed growth (Figure 4A). The present southerly distribution of parthenium weed could then be explained by this new scenario in which irrigation has been considered. When irrigation was added to the basic model under a climate change scenario (+3 °C), it was found that most of the southern parts of the Indus River basin in the Punjab province became suitable, while irrigation had no additional effects on distribution among the northern districts of the Sind province (Figure 4B). Similarly, the irrigation used will increase the suitability of land for parthenium weed in parts of the KP province including some irrigated areas of the Charsadda district.

### 2.2. Zygogramma bicolorata 

#### 2.2.1. Prediction under Current and Future Climates

The current distribution of the biological control agent for parthenium weed, *Z. bicolorata*, in Pakistan lies well within the CLIMEX projection model developed (Figure 5A). This model also shows that there are many areas suitable for the beetle’s growth outside of its present range. The model also suggests that the most suitable regions for the growth of the beetle are the northern districts of the Punjab province, while most of the southern Punjab, whole of Sindh and northern parts KP province seems to be unsuitable for the growth of *Z. bicolorata*. Under the climate change scenario (+3 °C), the potential distribution of the *Z. bicolorata* contracted within the Punjab province while it increased to cover more areas in the KP province (Figure 5B). The most suitable area for its growth seems to be in the northern extremities of the Punjab and KP provinces, as well as AJK, with an EI value of >30 (Figure 5B). Some areas in the Baluchistan province also became unsuitable under the climate change scenario.

#### 2.2.2. Prediction under Current and Future Climates with Irrigation

When the leaf-feeding beetle distribution model for the current climate was rerun with an irrigation function (0.5 mm day^−1^ winter and 1.0 mm day^−1^ summer) within the CLIMEX program, a significant part of the Indus River basin in southern Punjab and Sind became suitable for the growth of *Z. bicolorata* (Figure 6A). The model also showed that the irrigated parts on the Kabul and Kurram rivers in the KP province to become suitable for the growth of *Z. bicolorata* as well. These regions were previously predicted to be unsuitable for the growth of *Z. bicolorata* (Figure 4A and Figure 6A). The EI for *Z. bicolorata* decreased between the north and south in the Indus River basin except areas close to the Indus River delta, where the EI became suitable again for the Sind province. When the irrigation was added to the basic model again and then run under a climate change scenario (+3 °C), it was found that irrigation did not benefit *Z. bicolorata* and that its climatic suitability decreases across the whole of the Indus River basin (Figure 6B). This decrease in suitability under climate change is probably attributable to the beetle’s incapacity to survive under elevated temperatures (heat stress). The model suggested that only northern parts of the Punjab province and southern regions of the Sind province will remain suitable for the growth of the leaf-feeding beetle in the basin under climate change scenarios with the addition of irrigation (Figure 6B). The model also suggested that the irrigated areas on the Kabul River will also remain highly climatically suitable for the growth of the leaf-feeding beetle.

## 3. Materials and Methods

### 3.1. Data Collection

The geographical coordinates of 243 sites were recorded by the lead author to document the distribution of parthenium weed and *Z. bicolorata* in the Punjab province and ICT, Pakistan [9]. Geo-referenced data (88 points) on parthenium weed occurrence in KP province, Pakistan, was obtained from Dr Gul Hassan, professor at KP Agriculture University Peshawar, KP province. For Nepal, a total of 44 geo-referenced points for parthenium weed were acquired from Dr Bharat Babu Shrestha, Tribhuvan University Kathmandu, Nepal, while further parthenium weed occurrence points (551) in India, Pakistan, Bangladesh, and Sri Lanka were provided by Dr K. Dhileepan. These 551 points had been determined from a meta-analysis of published information, web searches, and opportunistic surveys. All points from the various sources were then geo-referenced through a popular mapping software Google Earth (http://earth.google.com accessed on 1 January 2012). Finally, all the data points for parthenium weed and *Z. bicolorata* were entered into spreadsheets and converted into respective layers of shapefiles compatible with a geographic information systems (GIS) mapping software (ArcMap version 10, Environmental Systems Resource Institute, Redlands, CA, USA).

### 3.2. The CLIMEX Model

CLIMEX is a dynamic model based on the weekly responses of a species to the climate estimated by calculating growth indices (GIs) and stress indices (SIs), which are in turn used to describe the potential growth of a population under favorable climatic conditions at a particular location [13]. Eight stress indices (cold, wet, hot, dry, cold-wet, cold-dry, hot-wet, and hot-dry) are used to predict the ability of a population to persist under unfavorable conditions. The stress indices estimate the threat to a species posed by prolonged or intensely extreme climatic conditions, and their interactions. The growth and stress indices are combined to calculate an ecoclimatic index (EI), which is then used as a measure of the favorability of a given locality for a species to grow. The EI values for locations range from 0 (unsuitable) to 100 (maximum suitability). In this study, the following EI classifications were used: EI = 0 (not suitable), EI = 1–15 (marginal), EI = 16–30 (suitable), and EI = > 30 (optimal). The “compare locations” function of the computer-based simulation package, CLIMEX (Version 3), was used to develop a distribution model for parthenium weed and *Z. bicolorata*. CLIMEX models are fitted based on either known distribution data or by available experimental inputs using an iterative parameter fitting process. This involves the manual adjustment of growth and stress parameters to develop a model and then compare it with the known distribution of a species. We used CM10-75H_V1.2mm climatic data in the CLIMEX modeling program.

#### 3.2.1. Parthenium Weed 

The original parameters of a predictive model for parthenium weed were developed by B. Lawson, as can be seen in unpublished data [3], using the CLIMEX semi-arid template. The parameters of the Lawson model and those modified by McConnachie et al. [3] were used as a baseline template for the current work. Since the southerly distribution of parthenium weed in Pakistan did not match the CLIMEX projection shown by McConnachie et al. [3], a new model was developed that considered these discrepancies by adjusting the temperature and moisture parameters (Table 1), and by later adjusting the irrigation patterns. This new model has been used to predict the current and potential distribution of the weed in Nepal [16].

The temperature and moisture indices were adjusted and fitted in accordance with the southern distribution limits of parthenium weed in Pakistan. The average summer temperatures in the southern districts of the Punjab province are in the range from 40 to 45°C (in this range for >60 days), with an annual precipitation of between 100 and 150 mm. The survival of parthenium weed in these regions suggests that the weed is comparatively more tolerant to drier and hotter conditions than previously considered. Parthenium weed can germinate within a wide range of temperatures (5–40°C), with a lower base temperature being estimated to be 7.2°C, while the temperature of 42.8 °C is predicted to be upper temperature threshold [25]. The high limiting temperature was adjusted from 39 to 42°C and the wet stress threshold values were decreased from 1 to 0.07. The soil moisture parameters were also adjusted based on the presence of this plant in a wide range of climates, from sub-humid tropical to hot-arid climates [26]. Furthermore, the same model was used to develop a distribution map for parthenium weed in south Asia using the climate change function of +3 °C. The change in current climatic suitability relative to climate change was determined by calculating the difference in EI values relative to both climates. 

An irrigation scenario (winter, 0.5 mm day^−1^, and summer, 1.0 mm day^−1^) was added to the basic CLIMEX model and was used to develop a predictive distribution model for parthenium weed under the current climate and under climate change scenarios.

#### 3.2.2. *Zygogramma bicolorata*

A CLIMEX model was developed based on data points collected in 2010 for the current distribution of *Z*. *bicolorata* and parthenium weed in Pakistan [9]. The original CLIMEX model used to predict the distribution of *Z. bicolorata* in south Asia [7] was based on meta data, and although it satisfies the current distribution of *Z. bicolorata* in Pakistan, it overestimates its potential geographic distribution in other parts of the world such as Australia. Keeping in mind the disagreement seen in the earlier model [7], a new model was developed based on the old model but with new distribution data [9] obtained from the current distribution study of *Z. bicolorata* in Pakistan (Table 2). The model has already been tested to predict the current and potential distribution of *Z. bicolorata* in Nepal [16]. 

A climate change (+3°C) and irrigation scenario (winter, 0.5 mm day^−1^, and summer, 1.0 mm day^−1^) using the CLIMEX program as described above were also developed to predict the potential future distribution of the *Z. bicolorata* in Pakistan. 

#### 3.2.3. General Model Development

All CLIMEX model data output files were imported into the ArcMap version 10 program (ESRI, Environmental Systems Resource Institute, Redlands, CA, USA) using relevant shapefiles, and from this, all the maps were developed. Specific data shapefiles for irrigated areas, rivers, and the canal system of Pakistan were obtained from the Worldwide Fund—Pakistan. Shapefiles of the countries along with their major administrative sub-divisions were downloaded from the DIVA-GIS website (http://www.diva-gis.org/ accessed on 1 January 2012). Specific shapefiles for the data on the distribution of parthenium weed and the leaf-feeding beetle in different countries were developed using the ArcMap 10 program.

## 4. Discussion

### 4.1. Potential Distribution of Parthenium Weed

The current climatic model predicts that the whole of South Asia is suitable for parthenium weed growth with the northern parts of Pakistan, India, Nepal, Sri Lanka, Bhutan, and Bangladesh being projected as being highly suitable. Parthenium weed has been confirmed to be present in all mainland countries of South Asia except Afghanistan, although many parts of Afghanistan were predicted as being suitable for parthenium weed growth, especially under a changing climate scenario (Figure 2). The present distribution records of parthenium weed in Pakistan, India, Nepal, and Bangladesh agree well with the predicted records. For example, parthenium weed is reported in all states of India with more frequent records of occurrence found in the southern states [7]. Parthenium weed is now considered an “up-and-coming weed” in Nepal and Bangladesh, and in Nepal, it is becoming a serious weed in protected areas such as the Chitwan National Park, home of the rare Asian one-horned rhinoceros (*Rhinoceros unicornis* L.). The model also satisfied the present distribution in Nepal and Bangladesh (data not shown). Using the same model as described here, Shrestha et al. [22] successfully predicted the current and potential distribution of parthenium weed in Nepal. 

In the late 1990s, parthenium weed was reported only in eight eastern districts of the Punjab province and ICT, Pakistan [10]. However, a later survey conducted in 2010 showed that the weed had spread to 28 of the 36 districts of the Punjab province [9]. Since then, it is likely that parthenium weed has invaded more regions in Pakistan. The climatic suitability to and spread of parthenium weed towards the cotton (*Gossypium hirsutum* L.)-growing areas in the Punjab and Sind provinces pose a grave threat to the cotton industry. Parthenium weed not only competes directly with this important crop, but it also acts as a secondary host to several insect pests (such as the cotton mealybug *Phenacoccus solenopsis* Tinsley) and important crop diseases (such as the tobacco streak virus; TSV) [27,28]. In Pakistan, TSV has caused severe yield losses in cotton crops [28], and the presence of parthenium weed in cotton-growing areas is now presenting the significant threat of further TSV outbreaks. 

A recent CLIMEX model developed by McConnachie et al. [3] for the prediction of the potential distribution of parthenium weed in eastern and southern Africa successfully predicted the present distribution of the weed in those regions, but when the same model was applied to Pakistan, some the weed’s presence in some of the southernmost regions were not predicted. These southern regions in which it was found, but not predicted, were within the Multan and Bahawalpur districts of the Punjab province. Climatically, these two districts are within one of the driest and hottest areas of Pakistan. The average rainfall in summer (April–August) is <150 mm and the average maximum temperatures from May to July are >42°C and up to 50°C. The survival of parthenium weed in these very hot and dry areas suggests that the weed could potentially survive under more extreme conditions than previously thought. In its native range, parthenium weed is found in a range of environments, ranging from humid subtropical to hot semiarid climates (e.g., in Mexico) [26], thus indicating that the weed has ability to tolerate a range of climatic conditions.

Pakistan has one of the largest canal irrigation systems in the world coming from the Indus River system that runs from the northeast to the southwest through the country. The Indus River, along with its major tributaries (Jhelum, Sutluj, Ravi, and Chenab) passes through the Punjab province and has created a huge fertile plain over many thousands of years. An extensive network of canals has been created along with various dams and barrages to irrigate the fertile plains of central and southern Punjab. When an irrigation scenario was added to the current CLIMEX model, most of the river basin became suitable for parthenium weed due to the availability of this extra moisture in the system. Under the irrigation scenario, these southern distribution points of parthenium weed, along with other southern regions in the Punjab and Sind provinces, became highly suitable for the growth of the weed. The western parts of the United States are considered climatically unsuitable for parthenium weed; however, some rare populations of the weed are found in agricultural areas under irrigation [26].

When the climate change scenario (+3 °C) was added to the current model, the potential range of parthenium weed contracted in South Asia. In South Asia, climate change will shift the spread of parthenium weed towards the north, while the southern regions will become less suitable (Figure 2). The model also suggests that parthenium weed can tolerate climate change conditions in most of the Indus basin due to the availability of the extra moisture in the form of irrigation.

### 4.2. Potential Distribution of Z. bicolorata

The distribution of *Z. bicolorata* and the predicted distribution by CLIMEX suggest that the climate in the northern regions of the Punjab province and some parts of the PK province is suitable for the beetle’s survival. Under climate change, the KP province will become more suitable to *Z. bicolorata*, a region where parthenium weed has already become a serious problem. Extension activities to redistribute the beetle among these new regions, where it is presently absent, are needed urgently. Similarly, some of the central districts of the Punjab province are climatically suitable to the beetle, and it would be beneficial to have the agent released there as well. 

The irrigation in the southern parts of the country will not only help parthenium weed to survive but will also help its biological control agent, *Z. bicolorata*, to survive. More areas of Pakistan would become favorable for the beetle under an irrigation scenario with extra moisture from irrigation, helping the beetle to survive in new areas. Irrigation will not help the leaf-feeding beetle to survive in some regions under climate change conditions (e.g., districts Sahiwal, Okara, and Khanewal) as these areas will become too hot for its survival.

*Zygogramma bicolorata* was first reported in a forest reserve near Lahore, Pakistan [23]. Presumably, this biological agent arrived in Pakistan from one of the releases made in northern India. The 2010 survey [9] revealed that *Z. bicolorata* is present in the northeast and northwest districts of the Punjab province but is entirely absent from the central and southern regions of the province [9]. A recent survey conducted a decade later, in the same districts, suggests that *Z. biolorata* is still restricted to the northern districts of Punjab; therefore, efforts may be required to redistribute it to other areas suitable for its growth [29].

## 5. Conclusions

Our study concludes that parthenium weed has the potential to spread to more locations within South Asia, particularly in Pakistan. Parthenium weed has already invaded some of the hot and drier locations in the southern parts of Pakistan that were previously not predicted to be suitable. Thus, there was a requirement to modify some of the environmental parameters of the CLIMEX model. Under climate change, parthenium weed is likely expand its distribution to the northern parts of most of the South Asian countries. Our modeling approach suggests that some areas are also suitable to one of its biological control agents (*Z. bicolorata*). The geographic range of this biological control agent in Pakistan can extend to other already infested parthenium weed areas in the southern parts of the Punjab province, into the KP province, and into the north of the country where this weed is likely to extend its range under future climatic conditions. This modeling approach also suggested that parthenium weed would be likely to spread to much larger areas that are under irrigation in the Indus basin. The biological control agent *Z. bicolorata* is likely to benefit from the irrigation in the southern areas of Pakistan. So, along with risks of further invasion of new areas, there are also opportunities for better planning and management of the invasive parthenium weed. 

## Figures and Tables

**Figure 1 plants-12-01381-f001:**
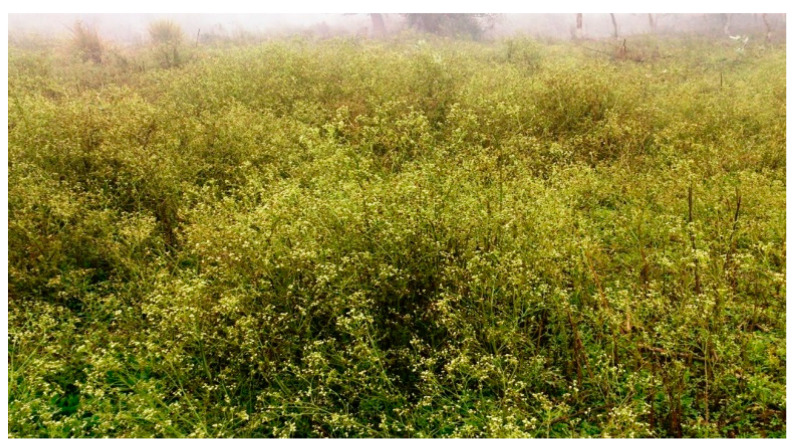
Parthenium weed invasion in Jhok reserve forest, Punjab, Pakistan.

**Figure 2 plants-12-01381-f002:**
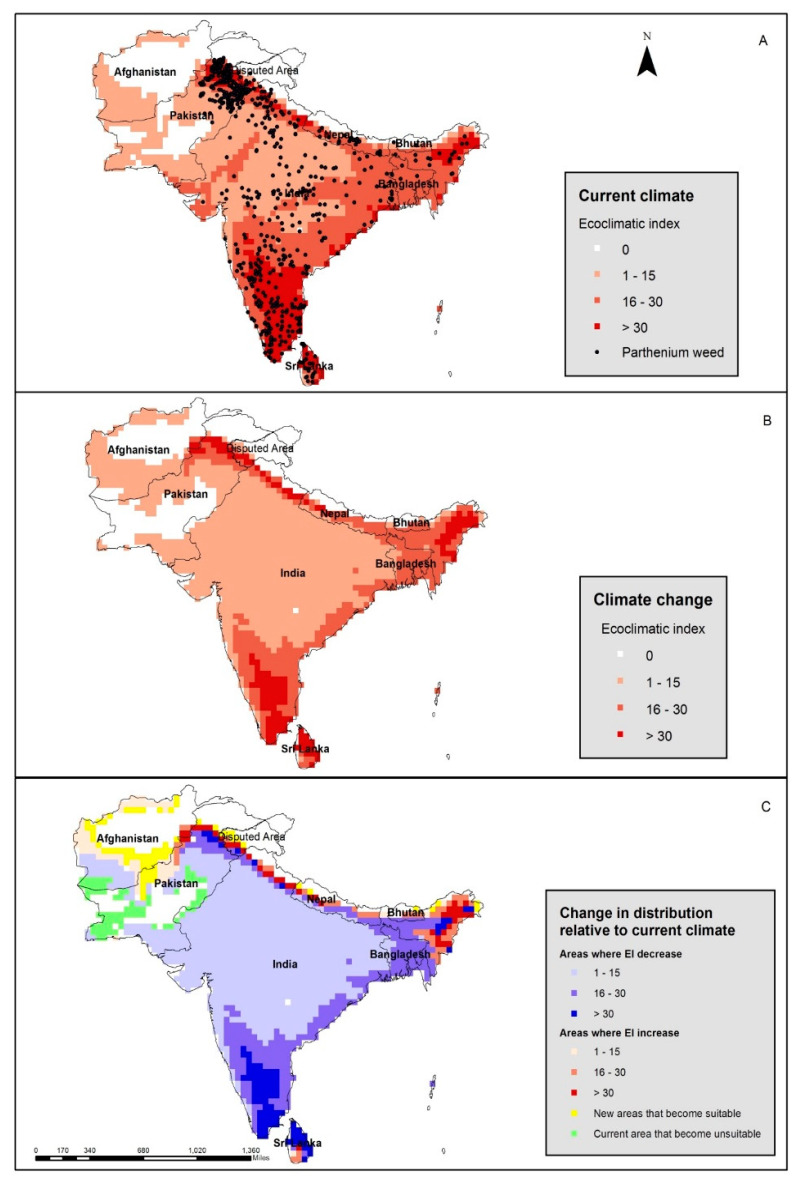
The current distribution of (black dots) and climatic suitability for parthenium weed in South Asia under (**A**) the current climate, and under (**B**) a changed climate with a change of +3 °C. The change in distribution, relative to the current climate (**C**), has been determined using the CLIMEX program.

**Figure 3 plants-12-01381-f003:**
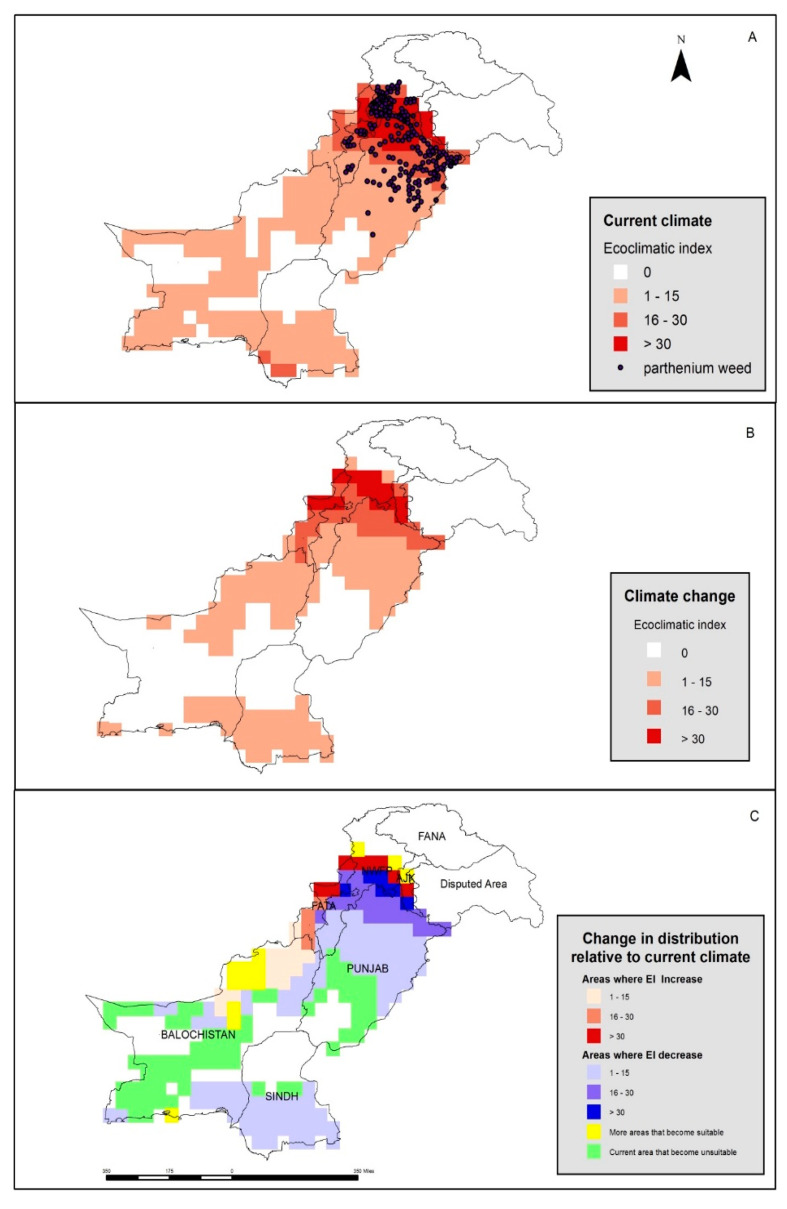
The current distribution of (black dots) and climatic suitability (colored squares) for parthenium weed in Pakistan under (**A**) the current climate, and under (**B**) a changed climate with a change of +3 °C. The change in distribution, relative to the current climate (**C**), has been determined using CLIMEX.

**Figure 4 plants-12-01381-f004:**
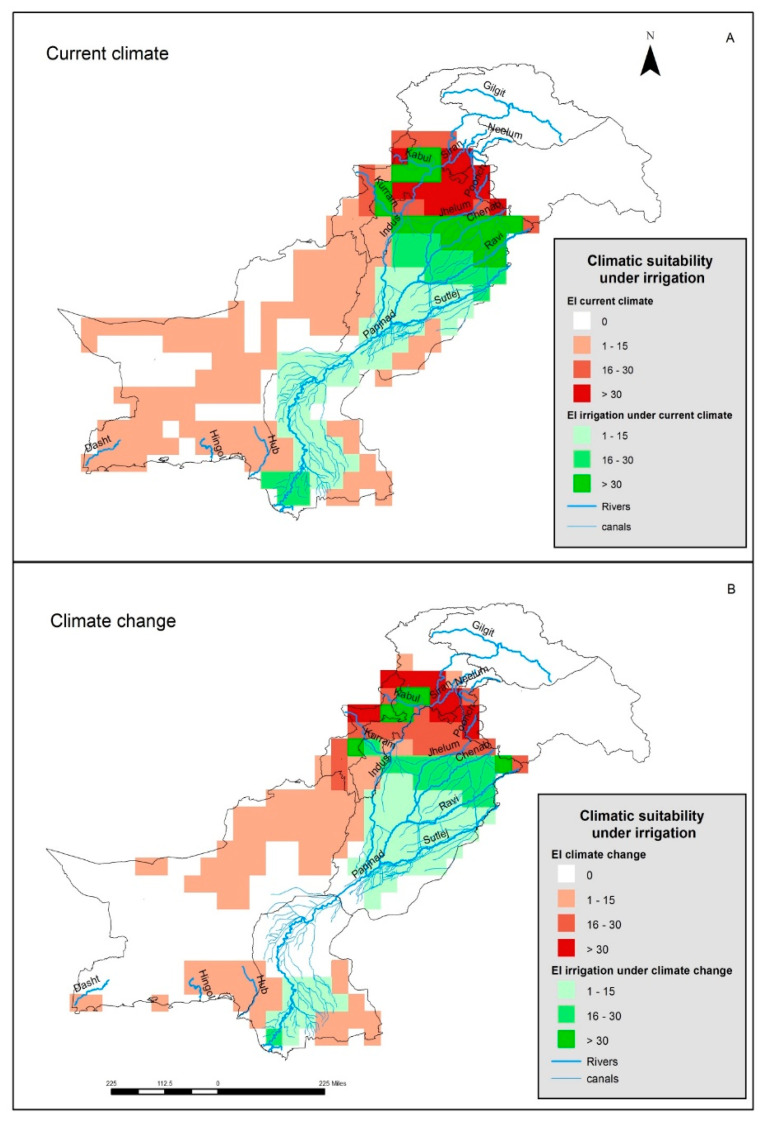
The climatic suitability for parthenium weed in Pakistan, when considering the extensive irrigation network, and (**A**) under the current climate or (**B**) under a climate change of +3 °C, modeled using CLIMEX.

**Figure 5 plants-12-01381-f005:**
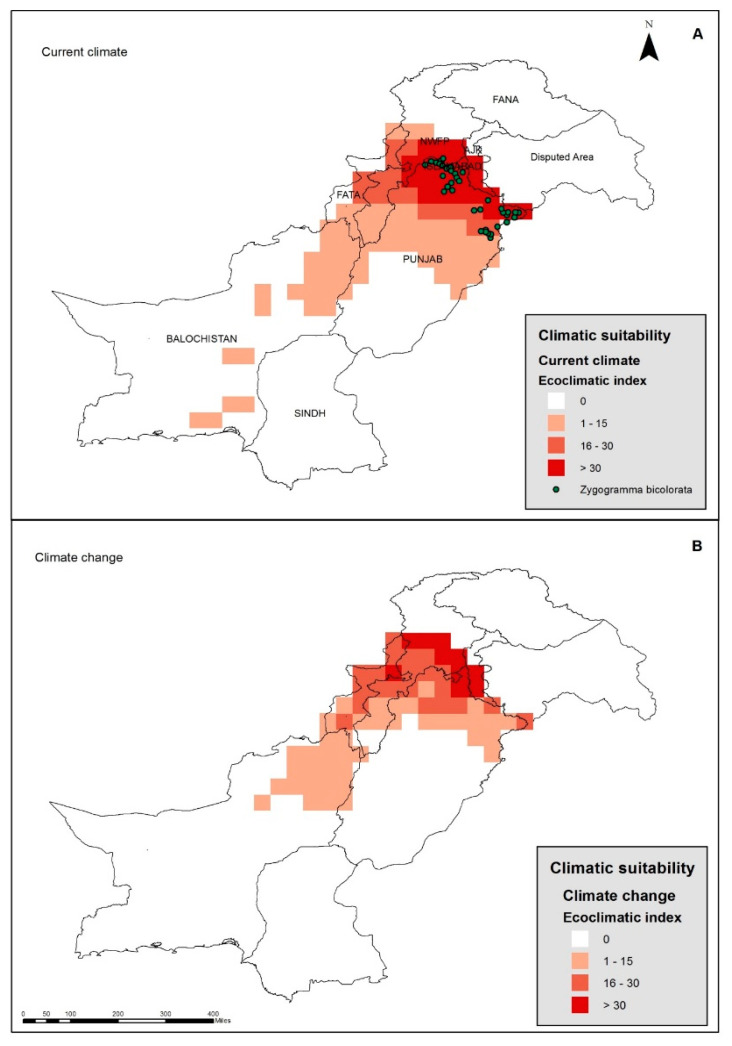
The current distribution of and climatic suitability for *Zygogramma bicolorata* in Pakistan under (**A**) the current climate and under (**B**) a changed climate with a change of +3 °C and modeled using CLIMEX.

**Figure 6 plants-12-01381-f006:**
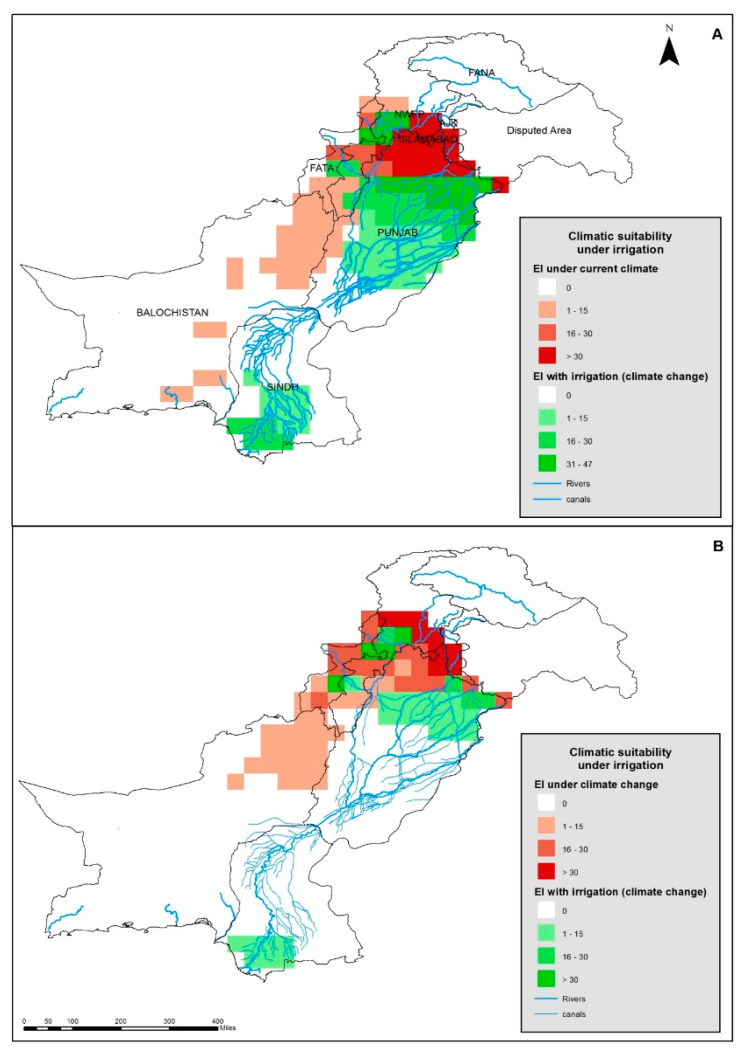
The climatic suitability for the *Zygogramma bicolorata* in Pakistan when considering the extensive irrigation network, and (**A**) under the current climate or (**B**) under a changed climate with a change of +3 °C, modeled using CLIMEX.

**Table 1 plants-12-01381-t001:** CLIMEX parameter values used in this present study and those of McConnachie et al. [3] for modeling the potential distribution of parthenium weed.

Index	Parameter	Values by McConnachie et al. [3]	Adjusted Values	Units
Temperature	DV0 = limiting low temperature DV1 = lower optimum temperature DV2 = upper optimum temperature DV3 = limiting high temperature	6 22 32 39	5 25 30 42	°C °C °C °C
Moisture	SM0 = limiting low soil moisture SM1 = lower optimum soil moisture SM2 = upper optimum soil moisture SM3 = limiting high soil moisture	0.1 0.3 0.8 1.4	0.08 0.2 0.6 1.6	
Cold stress	TTCS = temperature threshold THCS = stress accumulation rate DTCS = degree-day threshold DHCS = degree-day stress rate	4 −0.001 12 −0.0001	4 −0.001 12 −0.0001	°C week^−1^ day°C week^−1^
Heat stress	TTHS = temperature threshold THHS = stress accumulation rate	40 0.001	42 0.001	°C week^−1^
Dry stress	SMDS = wet stress threshold HDS = stress accumulation rate	0.1 −0.001	0.07 −0.001	week^−1^
Wet stress	SMWS = wet stress threshold HWS = stress accumulation rate	2.3 0.002	2.3 0.002	week^−1^
Hot-dry stress	TTHD = hot-dry temperature threshold MTHD = hot-dry moisture threshold PHD = stress accumulation rate	35 0.2 0.001	36 0.2 0.001	week^−1^
Annual heat sum	PDD = degree-day threshold	2000	2000	day°C

**Table 2 plants-12-01381-t002:** CLIMEX parameter values used in this present study and those of Dhileepan and Senaratne [7] for modeling the potential distribution of the biological control agent *Zygogramma bicolorata*.

Index	Parameter	Values Used by Dhileepan and Senaratne [7]	Adjusted Values	Units
Temperature	DV0 = limiting low temperature DV1 = lower optimum temperature DV2 = upper optimum temperature DV3 = limiting high temperature	12 18 35 40	12 18 32 39	°C °C °C °C
Moisture	SM0 = limiting low soil moisture SM1 = lower optimum soil moisture SM2 = upper optimum soil moisture SM3 = limiting high soil moisture	0.05 0.15 1.0 2.0	0.1 0.2 1.0 2.0	
Cold stress	TTCS = temperature threshold THCS = stress accumulation rate DTCS = degree-day threshold DHCS = degree-day stress rate	6 −0.01 7 −0.01	6 −0.001 7 −0.001	°C week^−1^ day°C week^−1^
Heat stress	TTHS = temperature threshold THHS = stress accumulation rate	40 0.005	40 0.005	°C week^−1^
Dry stress	SMDS = wet stress threshold HDS = stress accumulation rate	0.05 −0.05	0.05 −0.05	week^−1^
Wet stress	SMWS = wet stress threshold HWS = stress accumulation rate	2.0 0.005	2.0 0.005	week^−1^

## Data Availability

Not applicable.

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
