# Peer review of "The Current and Potential Distribution of Parthenium Weed and Its Biological Control Agent in Pakistan"

_plants, 2023, doi:10.3390/plants12061381_

Round 1

Reviewer 1 Report

Good paper with the future look on prediction and monitoring of specific weeds especially under climate changes and possibilities of irrigation (when water deficit is observed).

My suggestions are:

In yours figures are letters like A, B, C but in the text and under figures you used e.g. (Figure 1a or (a)). Use this same sings in whole paper.

Chapter 4. Materials and Methods please move in front of the paper after chapter 1. Introduction - (classic layout?!)

Author Response

Good paper with the future look on prediction and monitoring of specific weeds especially under climate changes and possibilities of irrigation (when water deficit is observed).

Response: Thank you very much for your encouraging comments, this is much appreciated.

My suggestions are:

In yours figures are letters like A, B, C but in the text and under figures you used e.g. (Figure 1a or (a)). Use this same sings in whole paper.

Response: Sure, I have modified all the figure references in the main text so that they are consistent with letters in figures. 

Chapter 4. Materials and Methods please move in front of the paper after chapter 1. Introduction - (classic layout?!)

Response: I have followed the template of the journal but, that's Ok, I have now moved this section (Chapter 4 Materials and Methods) after Chapter 1 (Introduction). The main headings and sub-headings are also modified accordingly.

Reviewer 2 Report

The article untitled The Current and Potential Distribution of Parthenium Weed and Its Biological Control Agent in Pakistan is in line with the aim and scope of Plants and may be published after a minor revision. Developing a predictive model of present and future distribution of invasive species under the climate change using modern modelling tool is novelty and important. An article is written very well, detailed remarks are listed below:

1. Introduction: The research goals are too general. There is a lack of tested hypotheses and pointed goals of the research.

2. The interpretation and discussion of the results should refer to the conditions of the current distribution of the species studied in their native range.

Author Response

The article untitled The Current and Potential Distribution of Parthenium Weed and Its Biological Control Agent in Pakistan is in line with the aim and scope of Plants and may be published after a minor revision. Developing a predictive model of present and future distribution of invasive species under the climate change using modern modelling tool is novelty and important. An article is written very well, detailed remarks are listed below:

Response: Thank you very much for your kind words, this is much appreciated.

Introduction: The research goals are too general. There is a lack of tested hypotheses and pointed goals of the research.

Response: We have modified the text of aims statements to make them more specific, please refer lines 97, 98 and 100 on page 3.

The interpretation and discussion of the results should refer to the conditions of the current distribution of the species studied in their native range.

Response: The information about native range distribution is added in discussion part (see lines: 342-245; 356-360)

Reviewer 3 Report

The timely and well-written response to the dual threats of climate change and invasive plants, with a useful prediction about biocontrol, is the type of paper we need right now. The methods could be clearer, but the message was clear and with a few edits, this could be a hard-hitting and highly useful paper.

Major Revisions
-Definitions of the stress indices in the CLIMEX model need precision.

- A more thorough explanation of the parameters for the predictive models for parthenium would help other researchers use the same techniques for their invasive plants of interest and would vastly increase the citation of this paper.

-Provide sources for where you got your climate data.

-For figures, putting goth the EI change and the areas that would become suitable and unsuitable on the same picture (like 1C and 2C) makes it difficult to see overlap. The newly suitable and unsuitable areas are the most interesting and the EI change is already present in parts A and B.

Minor Revisions

-A picture of parthenium weed would be nice.

-More information about where parthenium is native would build your story.

Line 45: the Wagah border

Line 136: Punjab Province

Line 169: "and" is crossed out but not removed

Line 367: "although" not needed

Line 380: "would likely to get benefit" mistranslated

Author Response

The timely and well-written response to the dual threats of climate change and invasive plants, with a useful prediction about biocontrol, is the type of paper we need right now. The methods could be clearer, but the message was clear and with a few edits, this could be a hard-hitting and highly useful paper.

Response: Thank you very much for your kind words and very useful suggestions. We have worked on the methods section to make it more clearer as suggested.  

Major Revisions
-Definitions of the stress indices in the CLIMEX model need precision.

Response: Stress indices are explained, see lines 124-125

- A more thorough explanation of the parameters for the predictive models for parthenium would help other researchers use the same techniques for their invasive plants of interest and would vastly increase the citation of this paper.

Response: We have explained the parameters used for parthenium weed model (refer lines: 151-157)

-Provide sources for where you got your climate data.

-For figures, putting goth the EI change and the areas that would become suitable and unsuitable on the same picture (like 1C and 2C) makes it difficult to see overlap. The newly suitable and unsuitable areas are the most interesting and the EI change is already present in parts A and B.

Response: Thanks for your comment, I understand the point regarding EI change is present in parts A and B but part C is important as it shows the intensity of increase and decrease in two different colour codes. With due respect, we would like to keep these figures as such. 

Minor Revisions

-A picture of parthenium weed would be nice.

Response: Ok, sure, we have added one as Figure 1.

-More information about where parthenium is native would build your story.

Response: Information about native range is built in first para of the Intro (ref: Line 38. 

Line 45: the Wagah border

Response: corrected.

Line 136: Punjab Province

Response: corrected.

Line 169: "and" is crossed out but not removed

Response: corrected.

Line 367: "although" not needed

Response: Ok, removed.

Line 380: "would likely to get benefit" mistranslated

Response: Corrected.